# Temperature Field and Stress Analysis of the Heavy-Concrete Transfer–Purge Chamber of the Nuclear Power Plant

**DOI:** 10.3390/ma16020613

**Published:** 2023-01-09

**Authors:** Xiaohui Wang, Xiaojun Li, Xuchen Liu, Yushi Wang, Aiwen Liu, Qiumei He, Chunlin Hou

**Affiliations:** 1Institute of Geophysics, China Earthquake Administration, Beijing 100081, China; 2Beijing Key Laboratory of Earthquake Engineering and Structural Retrofit, Beijing University of Technology, Beijing 100124, China; 3Nuclear and Radiation Safety Centre, Ministry of Environmental Protection of the People’s Republic of China, Beijing 100082, China

**Keywords:** transfer–purge chamber, heavy-concrete steel plate, temperature field, temperature stress

## Abstract

A transfer–purge chamber (TPC) is a double-steel-plate, heavy-concrete, curved-surface composite structure composed of steel plates, heavy concrete, and shear connectors. It is an important facility in the external refueling system of a nuclear power plant (NPP), providing a safe and reliable biological shielding space for reactor refueling operations. Temperature load is one of the most important factors that must be considered in the design of NPP structures. The temperature loads experienced by the TPC during its life cycle include those encountered in both normal and abnormal operation, which are distinct. In this study, we investigated the steady state and transient-state temperature fields and stresses of a TPC structure under normal operation and after 48 h of abnormal operation, respectively, which were calculated using Abaqus finite element software and the directly coupled method. During normal operation, the temperature field of the structure shows relatively uniform changes, and the temperature gradient of the internal concrete in the direction of its thickness has a constant value of 0.245 °C/cm. At the junction between the transfer and purge sub-chambers of the TPC, under the influence of wall curvature and deformation constraints, the maximum tensile strain of heavy concrete is 8.84 × 10^−3^, the maximum compressive strain is 2.04 × 10^−3^, the peak stress of the steel plate is 98.305 MPa, and the peak stress of the stud is 306.725 MPa. After 48 h of abnormal operation, the temperatures of the inner surface of the heavy concrete of the wall, the inner steel plate of the wall, the outer surface of the heavy concrete of the wall, and the inner steel plate of the wall increased by 8.12, 8.11, 0.31, and 0.30 °C, respectively. The tensile strain of the heavy concrete of the wall increased significantly by 52.64%, and the compressive strain of the concrete increased by 67.33%, whereas the stresses of the studs and steel plates increased by only 1.57% and 6.79%, respectively. These results show that the change in the temperature field greatly influences the stress and strain on the TPC structure. As measures for mitigating the development of this unfavorable situation of temperature stress concentration, the temperature operating range should be rationally controlled or the junction structure between the transfer and purge sub-chambers of the TPC optimized accordingly. The results of our study can provide basic data for a dynamic analysis of the TPC under conditions of combined earthquake and temperature loads.

## 1. Introduction

Temperature load is an important non-negligible load to be considered in the structural design of nuclear power plant (NPP) buildings, which are usually large structures operating at high temperatures, based on its potential to induce large structural stress. In the event of an accident involving transiently high temperatures, temperature-load-induced stress is a control factor that can potentially determine the bearing capacity of the structure [1]. The temperature loads that usually need to be considered for NPP building structures include the temperature effects of normal operation, LOCA(Loss of Coolant Accident) temperature effects, and temperature effects during severe accidents [2,3]. The temperature effect of NPP building structures during normal operation is the load borne in day-to-day operation and is a steady state temperature effect. The LOCA and severe accident temperature effects are transient temperature effects, i.e., the temperature field of the ambient temperature changes with time. The ambient temperature in the NPP building increases abnormally, and the temperature load on the structure also increases.

At present, research on the safety-related structures of NPPs mainly focuses on their ultimate bearing capacity. For example, the NUPEC (Nuclear Power Engineering Corporation) and the NRC (US Nuclear Regulatory Commission) jointly conducted more systematic and comprehensive research on the ultimate bearing capacity of a nuclear reactor containment under internal pressure [4,5]. On the basis of other research, Wang et al. [6] performed a structural test analysis of NPPs on non-rock sites. Much research has been conducted on the safety-related structures of NPPs under the effect of temperature. Qian et al. [7] analyzed the temperature fields of three typical construction layers of a concrete containment using Abaqus finite element (FE) analysis software and comparatively analyzed the effects of the construction layer height and thickness, pore geometry and size, thermal insulation performance, and curing time on the temperature field of the structure. Lin and Yan [8] performed a time-history analysis and quantitative estimation of the temperature field problem of NPP containment under an atmospheric environment. Li and Li [9] used the FE method to analyze the steady- and transient-state temperature fields of a pressurized water reactor NPP containment during normal operation and severe accidents, respectively. They simulated the temperature field variations in the containment to determine the most severe temperature effect. Wu et al. [10] obtained the temperature field distributions in a containment wall at different moments of a LOCA using the heat transfer analysis method, which led to the proposal of a simplified method for analysis of the internal force of containment structures under the temperature effect of a LOCA based on elastic mechanics theory. Zhang [11] simulated and calculated the distribution of the temperature field in the overall sealing test of the NPP containment. The change in the temperature field in the containment and the flow state of the airflow were given, and the phenomenon of temperature stratification distribution in the space 34 m above the containment was verified. Ye et al. [12] used three-dimensional solid elements to establish a finite element model of a steel-plate–concrete modular wall, on which they performed transient thermal and thermal stress analyses. They proposed a calculation method for the bearing capacity of steel plate and concrete while considering temperature changes during an accident, thereafter, improving the design of shear studs, angle steel, and channel steel. The mode of connection between the wall and the surrounding components was discussed to overcome the influence of the thermal effect. Sun et al. [13] studied the fire resistance and nuclear safety of steel-plate–concrete composite shear wall structures. The influence of factors such as wall thickness, steel plate thickness, stud spacing, and height-to-thickness ratio on the fire resistance of eight specimens was analyzed through integral tests.

Compared with the reinforced concrete structure used in traditional safety-related structures of an NPP, a steel-plate, heavy concrete structure is adopted for TPCs [14]. This structure is a thick, arc-shaped wall with a cantilever. It has the characteristics of thick components, a long cantilever, and complex structural forces. Steel plates, connected by studs, are located on the inside and outside of the heavy concrete wall. The heat resistance and thermal expansion coefficient of steel plates and studs differ from those of heavy concrete, and the temperature resistance is not as good as for reinforced concrete structures. In addition, the accident-based design of the TPC ensures it will withstand a certain temperature effect in such a scenario. Therefore, in addition to conducting research on the conventional seismic performance and mechanical performance of steel-plate heavy-concrete arc wall structures, it is also extremely necessary to conduct in-depth research on their structural performance under transient temperature loads, such as those encountered during accidents.

## 2. Basic Theory of Heat Transfer

In thermodynamics, there are three modes of heat transfer: conduction, convection and radiation [15]. Under normal operating conditions, the temperature field of the TPC remains unchanged, and the effect of the temperature load on the structure is in a steady state, thus allowing for its steady state analysis. Under abnormal operating conditions, the temperature field of the TPC experiences temperature increases from the normal steady state temperature field and features transient temperature variations, thus enabling transient-state analysis of the process.

The internal and external ambient environments of a TPC structure are uniform, have stable air, and have flat and smooth inner and outer surfaces. The temperature field of the TPC structure can be approximately considered as varying uniformly in the direction of the wall thickness, whereas thermal conduction can be considered a one-dimensional (1D) thermal conduction problem [16]. For the steady state 1D thermal conduction problem, the temperature gradient in the flat wall is constant [10], and the thermal conduction is a dynamically stable process. Figure 1 illustrates the process of heat transfer in a TPC.

Here, T_1_ and T_2_ are the air temperatures outside and inside the TPC structure, respectively; T_w1_ and T_w2_ are the temperatures on the outer and inner surfaces of the TPC wall, respectively; Q_1_ is the volume of conduction heat transfer from the outer surface of the TPC wall to the external air; Q_2_ is the volume of heat transfer from the inner surface to the outer surface of the TPC wall; and Q_3_ is the volume of heat transfer from the air inside the TPC to the inner surface of the TPC wall. A_1_ and A_2_ are the air-exposed areas of the outer and inner surfaces of the TPC. Since the diameter of the TPC structure is large and the diameter and height are much larger than the thickness of the wall, the calculations can be conducted according to the heat conduction problem of a large flat wall in heat transfer, that is, $A_{1}=A_{2}=A$. h_1_ and h_2_ are the convection heat transfer coefficients between the outer and inner surfaces of the TPC and air, respectively, and λ and δ are the equivalent thermal conductivity and thickness of the TPC wall and ceiling, respectively.
(1)$$Q_{1}={\mathrm{Ah}}_{1}\left(T_{w1}-T_{1}\right),$$
(2)$$Q_{2}=\frac{A\lambda }{\delta }\left(T_{w2}-T_{w1}\right),$$
(3)$$Q_{3}={\mathrm{Ah}}_{2}\left(T_{2}-T_{w2}\right),$$

Combining Equations (1)–(3) results in the elimination of the temperatures Tw_1_ and Tw_2_ on both sides of the wall; then, the heat flow ($Q_{31}$) from the internal environment of the structure to the atmospheric environment can be expressed as follows:(4)$$Q_{31}=\mathrm{AK}\left(T_{2}-T_{1}\right)$$
(5)$$K=\frac{1}{\frac{1}{h_{1}}+\frac{1}{h_{2}}+\frac{\delta }{\lambda }}$$
where K is the heat transfer coefficient, which is equal to the value of the heat flow rate when the heat transfer area equals A = 1 m^2^ and when the temperature difference between the cold and hot fluids is ∆t = 1 °C. It is a scale that characterizes the intensity of the heat-transfer process. The stronger the heat-transfer process, the larger the heat-transfer coefficient, and vice versa. For components with known shape, size, and wall thickness, K and A are constants in Formula (4). The heat flow Q_31_ is related to the temperature outside and inside the TPC structure.

When the temperature field is in a steady state,
(6)$$Q_{1}=Q_{2}=Q_{3}$$

According to the Fourier thermal conduction equation, the temperature conduction in the TPC structure can be expressed as [10,17]
(7)$$\rho c\frac{\partial T}{\partial \tau }=\lambda \left(\frac{\partial^{2}T}{\partial x^{2}}+\frac{\partial^{2}T}{\partial y^{2}}+\frac{\partial^{2}T}{\partial z^{2}}\right)$$

In this research, the heat conduction can be equivalent to a one-dimensional heat conduction problem along the thickness direction of the wall, and Formula (7) can be written as:(8)$$\rho c\frac{\partial T}{\partial \tau }=\lambda \frac{\partial^{2}T}{\partial d^{2}}$$
where T is the function for the non-steady state temperature field variation in the TPC, τ is the time variable, ρ and c are the density and specific heat capacity of the TPC wall and ceiling, respectively. The thermal conduction of the transient-state temperature field of the TPC is unstable, i.e., Q_1_, Q_2_, and Q_3_ are nonequal terms.

## 3. FE Model

In this paper, a finite element model is developed based on the transfer and purging chamber of a fourth-generation nuclear power plant in China. The transfer and purging chambers are mainly composed of a transfer chamber and a purging chamber. The transfer and purging chambers are located on the 20.20 m elevation floor of the NPP containment, with an elevation of 20.20−31.90 m. The inner and outer diameters of the transfer chamber are 4100 and 5100 mm, respectively, and the wall thickness is 1004 mm. The inner and outer diameters of the purge chamber are 3750 and 4750 mm, respectively, and the wall thickness is 1004 mm. The wall is a double-layer steel plate–concrete structure; the thickness of the inner and outer steel plates is 22 mm, and the thickness of the concrete is 960 mm. 

In order to ensure the consistency of the structural calculation and analysis model of the TPC under temperature load, earthquake load, and combined temperature load and earthquake loads, the mesh size of the finite element model used in this paper is kept consistent with that used in the seismic fragility analysis by Liu et al. [18]. Liu et al. conducted a sensitivity analysis on the mesh of the TPC finite element model and showed that when the mesh size of the transfer cleaning room is 500 mm, the first natural vibration frequency error is small, and computing resources are saved. Thus, a mesh size of 500 mm was used in the model in this study. In selecting the unit type of the finite element model, both the heat transfer and the force transfer should be considered, so the C3D8T unit is adopted for the concrete, the S4RT unit is adopted for the steel plates, and the T3D2T unit is adopted for the studs. The location and the finite element model of the TPC are shown in Figure 2.

### 3.1. Material Parameters

The primary building materials of the TPC include C40 hematite concrete, Q345 steel plates, and SL15 studs(Beijing Baoheyuan Equipment Co., Ltd., Beijing, China). The material parameters required for temperature field analysis are as follows:(1)The C40 hematite concrete has a modulus of elasticity of 50,000 MPa [19], a linear expansion coefficient of 1.0 × 10^−5^/°C [20], a thermal conductivity of 2.85 W/(m∙k), a specific heat capacity of 912 J/(kg∙k) [21], and an initial temperature of 20 °C.(2)The steel plates and studs have a modulus of elasticity of 102,000 MPa, a linear expansion coefficient of 1.2 × 10^−5^/°C [20], a thermal conductivity of 36.7 W/(m∙k), a specific heat capacity of 531 J/(kg∙k) [21], and an initial temperature of 20 °C. Table 1 shows the mechanical and thermal parameters of the materials.

The temperature range considered in this study is 5–70 °C. The mechanical and thermal parameters vary insignificantly within this temperature range; thus, the effects of temperature on the material parameters were neglected.

### 3.2. Boundary Conditions and Constraints

The indoor air temperature of the TPC is 50 °C during normal operation or shutdown and 70 °C during an accident, which is defined as lasting for 48 h. At these temperatures, the effects of the heat radiation of the structure are insignificant [14] and can thus be neglected. As a result, only heat transfer and convection were considered.

The concrete of the TPC is closely bonded with the inner and outer steel plates, and there is continuous temperature and heat flow at the interface. Thus, the interface is defined as a full-contact boundary condition [22], i.e., the two contacting surfaces are bonded, with no thermal contact resistance at the interface, enabling free heat transfer. According to the design requirements, the ambient temperatures inside and outside the TPC are known: 50 and 5 °C during normal operation or shutdown and 70 and 5 °C during an accident, respectively. According to the principle of heat transfer, the temperature boundary condition of the TPC structure is a third-type boundary condition [1], i.e., the air temperatures inside and outside the TPC structure are stable, and the heat flux density at the air–structure interface (i.e., the surface heat-transfer coefficient) is constant. The convection heat-transfer coefficients between the concrete and indoor air, between the concrete and external air, between the steel plate and indoor air, and between the steel plate and external air were set to 8, 16, 4.65, and 16.27 W/m^2^∙°C, respectively [23].

The mechanical boundary conditions for the temperature field analysis of the TPC were defined as follows. The degree of freedom in the vertical displacement of the concrete ground of the structure was constrained, with free deformations allowed for other degrees of freedom.

## 4. Steady State Temperature Field under Normal Operation

### 4.1. Temperature Field

In the normal-operation state of the TPC, it is considered that a steady state has been reached after adequate heat transfer. The temperature field and stress are basically stable. Figure 3 shows the temperature fields of the individual components of the steady state structure.

The temperature field of the steel plates is characterized by a uniform distribution. During normal operation, the inner and outer steel plates of the structure have maximum surface temperatures of 35.475 °C and 6.48 °C, respectively. The temperature field of the internal concrete is characterized by a uniform distribution. During normal operation, the inner- and outer-surface temperatures of the concrete are 35.475 and 6.441 °C, respectively. Figure 4 shows the temperature gradient of the concrete in the direction of its thickness. As shown in the figure, the temperature gradient of the heavy concrete in the thickness direction has a constant value of 0.245 °C/cm.

### 4.2. Temperature Stress

Figure 5 shows the temperature stresses of the concrete, steel plates, and studs. As shown in the figure, under a temperature load corresponding to normal operation, the heavy concrete of the TPC structure is generally in compression and locally in tension. The maximum tensile and compressive strains of the heavy concrete exhibit “local highs”. In other words, the compressive and tensile strain responses of the heavy concrete are generally small; however, at the junction between the transfer and purge sub-chambers, the compressive and tensile strength responses of the concrete are markedly excessive. Owing to the complex cross-section, large curvature of the wall, and numerous constraints on the deformation, the tensile and compressive strains of the heavy concrete in this region are relatively large under the action of the temperature load; the maximum tensile strain is 8.84 × 10^−3^ and the maximum compressive strain is 2.04 × 10^−3^, which exceeded the corresponding values for the ultimate compressive strain and tensile strain of heavy concrete, which are 1.45 × 10^−3^ and 0.15 × 10^−3^, respectively. Similarly, the stress responses of the steel plates and studs are significantly higher in this region than in other regions. In particular, the peak stress of the steel plate is 98.305 MPa, which is far below its yield strength of 355 MPa; the peak stress of the stud is 306.725 MPa, which has exceeded its yield strength of 270 MPa, whereas the stress of the studs in other regions was extremely small. 

## 5. Transient-State Temperature Field under Abnormal Operation

### 5.1. Temperature Field

During abnormal operation, the internal ambient temperature of the TPC increased from 50 to 70 °C, and the temperature field of the structure was not stable. The transient-state temperature fields in selected regions of the TPC structure were extracted. Table 2 shows the temperature field variations in the TPC structure after 48 h of abnormal operation.

As shown in Table 2, and the location of nodes A–J are shown in Figure 6,the temperature field of the TPC varies under abnormal operation, and the temperature of the structure increases. After 48 h of abnormal operation, the inner-surface temperature of the TPC structure is markedly increased. The temperatures of the concrete and steel plates increase to differing extents. After 48 h of abnormal operation, the temperatures of the inner surfaces of the steel-bar reinforced concrete ceiling, the inner surface of the heavy concrete of the wall, the inner steel plate of the wall, the outer surface of the heavy concrete of the wall, and the inner steel plate of the wall increased by 11.64, 8.12, 8.11, 0.31, and 0.30 °C, respectively. Figure 7 shows the temperature variations in the TPC structure with time under abnormal operation.

As shown in Figure 7, the inner surface temperature of the TPC increases after operating under abnormal conditions. The temperatures of the inner steel plate and the inner concrete surface of the TPC were similar, with a minimal temperature difference. The temperatures of the inner steel plate and the inner concrete surface were approximately 40.8 °C, markedly lower than that of the structured ceiling at approximately 50.9 °C. At the beginning of the abnormal operation, the temperature inside the TPC structure increases rapidly. As the structural temperature increases, the rate of increase in the temperature of the structure gradually slows down, but the temperature continues to increase.

### 5.2. Temperature Stress

Table 3 shows the temperature stress and strain of the TPC after 48 h of abnormal operation. As shown in the table, the stress and strain on the TPC increased after 10 h of abnormal operation. In particular, the tensile strain of the heavy concrete of the wall increases significantly by 52.64%, and the compressive strain of the concrete increases by 67.33%. The strains of the steel plates and studs were not markedly increased. The stresses of the studs and steel plates increased by only 1.57% and 6.79%, respectively, and the tensile and compressive strains of the concrete ceiling increased by 47.35% and 42.43%, respectively. These results indicate that, under abnormal operation, the temperature increases inside the TPC structure, leading to a significant increase in the temperature stress of the structure. In particular, the strain varies more significantly for the concrete of the TPC than for the steel plates and studs. The concrete strain is therefore more sensitive to temperature.

## 6. Conclusions

In this study, steady- and transient-state analyses of the heavy-concrete TPC of NPPs under normal operation and after 48 h of abnormal operation, respectively, were performed. The major conclusions are summarized as follows:(1)The effects of the temperature load are non-negligible in the design and operation of NPPs, and a temperature load could lead to the concrete cracking and the studs yielding in the TPC structure.(2)Under normal operation, the steady state temperature field significantly affects the stress and strain of the TPC structure, especially for the concrete at the TPC junction.(3)During normal operation, the temperature field of the structure changes relatively uniformly, and the temperature gradient along the direction of thickness is 0.245 ℃/cm. Regarding local changes at the connection between the transfer chamber and the cleaning chamber, the maximum tensile strain of the heavy concrete is 8.84 × 10^−3^, the maximum compressive strain is 2.04 × 10^−3^, the peak stress of the steel plate is 98.305 MPa, and the peak stress of the stud is 306.725 MPa due to the influence of wall curvature and deformation constraints.(4)After 48 h of abnormal operation, the temperatures of the inner surface of the heavy concrete of the wall, the inner steel plate of the wall, the outer surface of the heavy concrete of the wall, and the inner steel plate of the wall increase by 8.12, 8.11, 0.31, and 0.30 °C, respectively. The tensile strain of the heavy concrete of the wall increased significantly by 52.64%, and the compressive strain of the concrete increased by 67.33%. The stresses of the studs and steel plates increased by only 1.57% and 6.79%, respectively. Compared with normal operating conditions, there are significantly greater stress–strain increments for concrete than for steel plates and studs under abnormal operating conditions.(5)The distribution of stress and strain in the TPC structure is uneven under the action of temperature load, and there is a relatively serious phenomenon of temperature stress concentration. Affected by the large local curvature of the outer surface of the structure, the stress or strain levels of the outer steel plates, studs, and concrete at the connection between the transfer chamber and the cleaning chamber are even higher than those of other parts. Optimizing this junction or taking other measures to eliminate the temperature stress concentration in this part is recommended.

## Figures and Tables

**Figure 1 materials-16-00613-f001:**
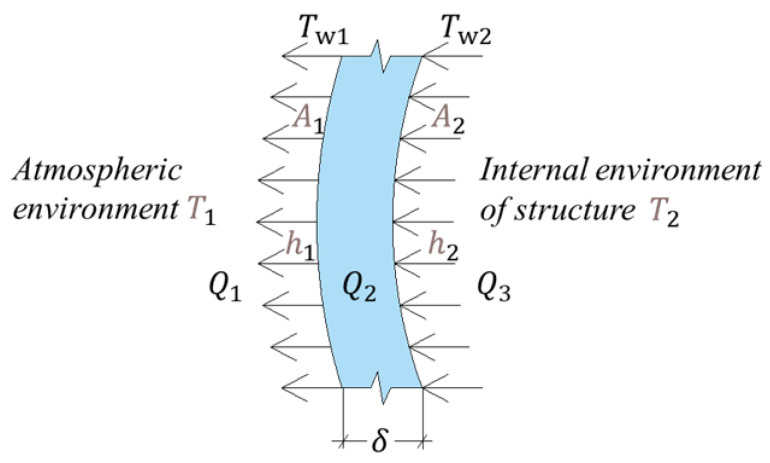
Illustration of heat transfer in a TPC. (The arrow direction is the temperature transmission direction).

**Figure 2 materials-16-00613-f002:**
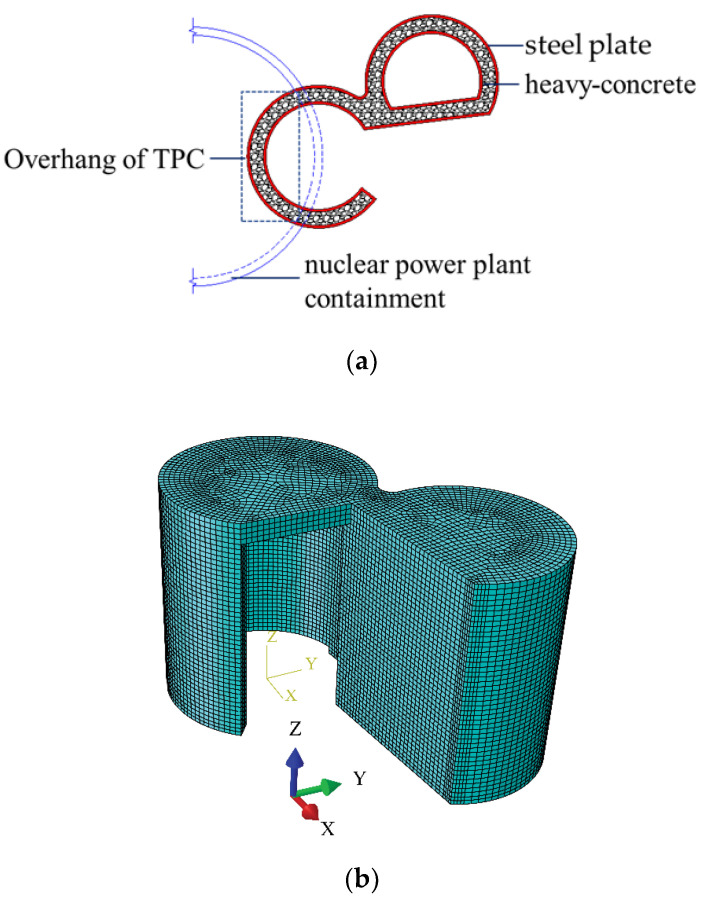
The location and the finite element model of the TPC. (**a**) The location of TPC. (**b**) The finite element model of the TPC.

**Figure 3 materials-16-00613-f003:**
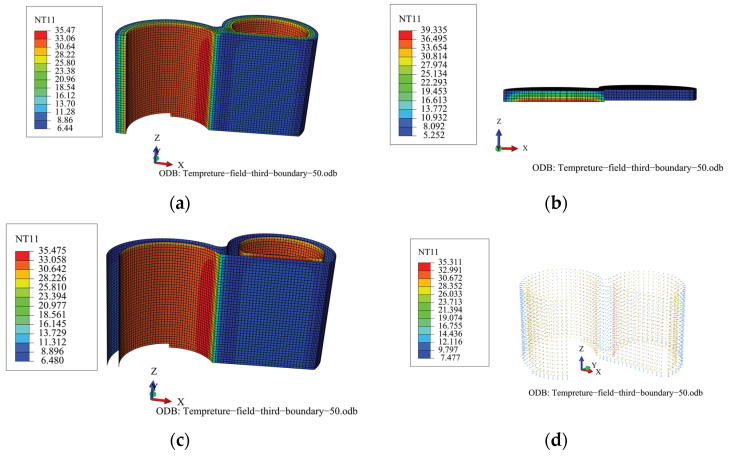
Temperature field of the TPC during normal operation. (**a**) Internal concrete; (**b**) Concrete roof; (**c**) Inner and outer steel plates; and (**d**) Stud.

**Figure 4 materials-16-00613-f004:**
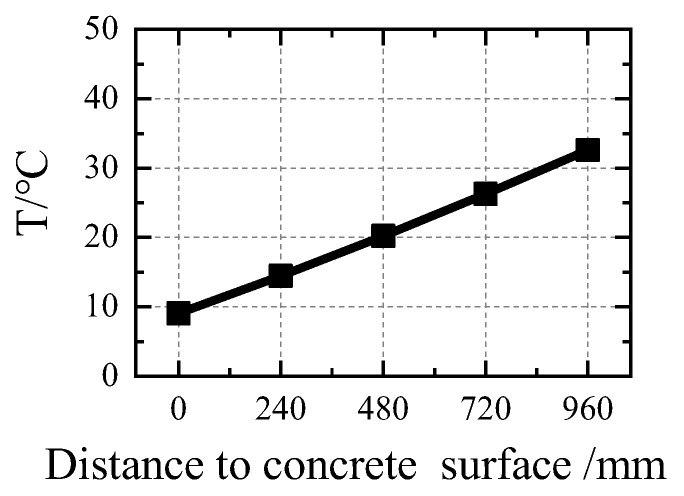
The temperature field of the internal concrete under normal operation.

**Figure 5 materials-16-00613-f005:**
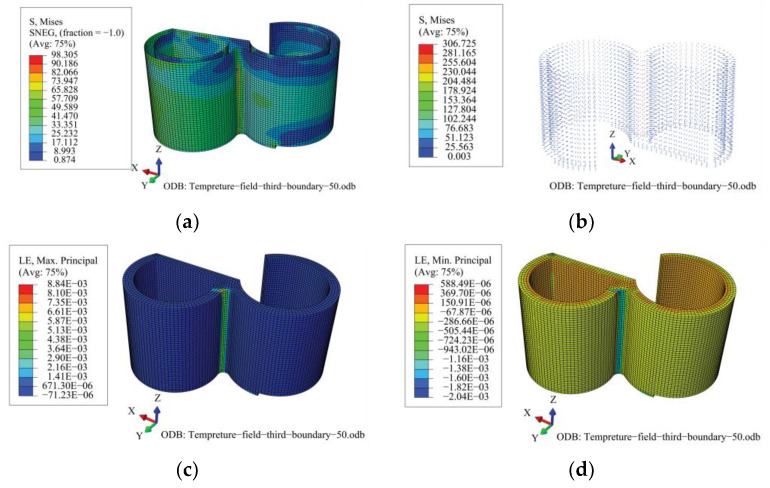
Temperature stress or strain in the TPC. (**a**) Stress of steel plates; (**b**) Stress of studs; (**c**) Tensile strain of concrete; and (**d**) Compressive strain of concrete.

**Figure 6 materials-16-00613-f006:**
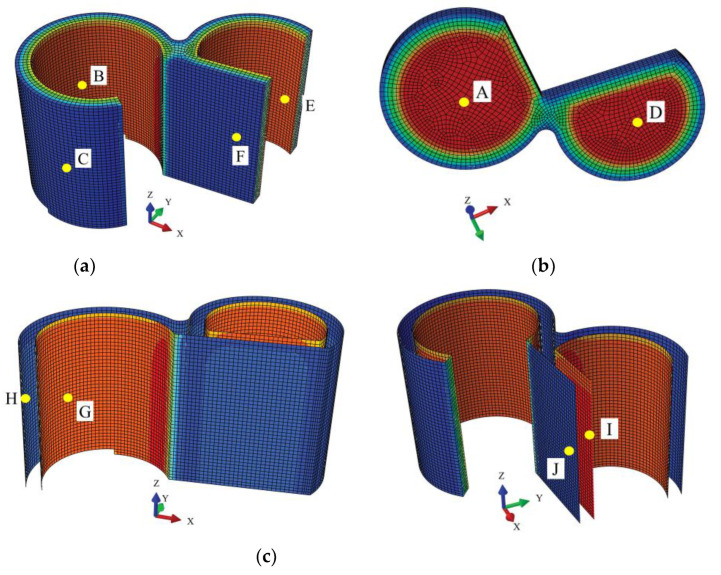
Schematic diagram of a temperature field node in the TPC. (**a**) Surface of transfer sub-chamber wall; (**b**) Surface of transfer sub-chamber ceiling; and (**c**) Steel plate of transfer sub-chamber.

**Figure 7 materials-16-00613-f007:**
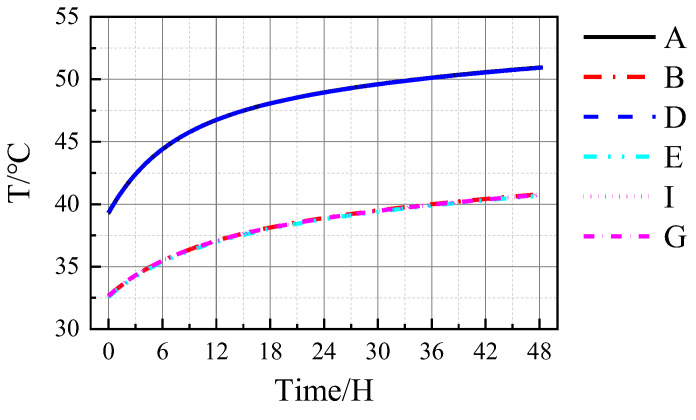
Transient temperature field inside the TPC.

**Table 1 materials-16-00613-t001:** Mechanical and thermal parameters of materials.

Parameter	C40 Hematite Concrete	Q355 Steel Plates	ML15 Studs
Modulus of elasticity (MPa)	50,000	206,000	210,000
Poisson’s ratio	0.30	0.2	0.2
Density (kg·m^−3^)	3700	7850	7850
Yield strength (MPa)	19.1 (compressive) 2.01 (tensile)	345	270
Linear expansion coefficient (°C^−1^)	1.0 × 10^−5^	1.2 × 10^−5^	1.2 × 10^−5^
Thermal conductivity [W/(m∙k)]	2.85	36.7	36.7
Specific heat capacity [J/(kg∙k)]	912	531	531

**Table 2 materials-16-00613-t002:** Variation in the temperature field of the TPC during abnormal operation. (The location of nodes A–J are shown in Figure 6).

Component	Designation of Serial Node	Location of Node	Temperature under Normal Operation (°C)	Temperature after 48 h of Abnormal Operation (°C)	Temperature Variation (°C)
Concrete	A	Inner surface of transfer sub-chamber ceiling	39.33	50.96	11.63
	B	Inner surface of transfer sub-chamber wall	32.68	40.81	8.13
	C	Outer surface of transfer sub-chamber wall	9.06	9.37	0.31
	D	Inner surface of purge sub-chamber ceiling	39.29	50.94	11.65
	E	Inner surface of purge sub-chamber wall	32.59	40.70	8.11
	F	Outer surface of purge sub-chamber wall	9.01	9.32	0.31
Steel plate	G	Inner steel plate of transfer sub-chamber	32.65	40.76	8.11
	H	Outer steel plate of transfer sub-chamber	9.02	9.33	0.31
	I	Inner steel plate of purge sub-chamber	32.65	40.76	8.11
	J	Outer steel plate of purge sub-chamber	8.98	9.28	0.30

**Table 3 materials-16-00613-t003:** Temperature stress and strain of the TPC under abnormal operation.

Item	Normal Operation	After 48 h of Abnormal Operation	Ratio of Increase
Maximum tensile strain of heavy concrete of the wall	$8.91\times 10^{-3}$	$13.6\times 10^{-3}$	52.64%
Maximum compressive strain of heavy concrete of the wall	$2.02\times 10^{-3}$	$3.38\times 10^{-3}$	67.33%
Maximum von Mises stress of steel plates	98.98 MPa	105.70 MPa	6.79%
Maximum von Mises stress of studs	306.61 MPa	311.42 MPa	1.57%
Maximum tensile strain of the concrete ceiling	$69.70\times 10^{-3}$	$102.7\times 10^{-3}$	47.35%
Maximum compressive strain of concrete ceiling	$23.12\times 10^{-3}$	$32.93\times 10^{-3}$	42.43%

## Data Availability

Not applicable.

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
