# Peer review of "Temperature Field and Stress Analysis of the Heavy-Concrete Transfer–Purge Chamber of the Nuclear Power Plant"

_materials, 2023, doi:10.3390/ma16020613_

Round 1
Reviewer 1 Report
Comments
The paper aims to investigate the steady- and transient-state temperature fields and stresses of the TPC structure, under normal operation and after 48 h of abnormal operation, by using the Abaqus finite-element software and the directly coupled method. However, major modification should carry out before publishing as follows:
1) In the abstract, the author stated "after 48 h of abnormal operation, the inner surface temperature of the TPC increased markedly". Please clarify the increment percentages and revised the abstract to highlight the novelty of the study.
2) The introduction must improve and additional references should add to highlight the problem statement and purpose of the current study. The results of prior studies must add to reflect the effect of temperature load on the nuclear power plant buildings.
3) The information about the finite element model is poor and additional information must add regarding the adopted geometry and mesh size and the convergency test.
4) How did you validate the purpose model? Please clarify
5) The results are poor and not sufficient to reflect the effect of temperature load on the behavior of NPP structures.
6) Extensive English editing is recommended.

Author Response
Dear reviewer,
Thank you very much for your review on 11 Nov. 2022 with which you sent us the yours suggestions on our paper with the reference number materials-2007156. We also wish to take this opportunity to thank you for his constructive comments and valuable recommendations. We have carefully revised the manuscript according to your suggestion. The following reply hopes to answer your review, The following reply hopes to answer your review,Please see the attachment.
If the reply cannot explain your suggestion, I hope to get your guidance again, thank you very much !
Best wishes!
Wang xiaohui

Reviewer 2 Report
Dear Editor, Dear Authors
Paper contains an analysis of the state of stresses and deformations at temperature loads experienced by the transfer-purge chambers of nuclear power plant during their life cycles include the temperature loads under normal and abnormal operation.
Theoretical analysis was performed for the assumed mechanical and thermal parameters of materials. The temperature range considered in this study was 5–70 ° C. The mechanical and thermal parameters vary insignificantly within this temperature range; thus, the effects of temperature on the material parameters were neglected.
Remarks:
1. Limiting the analysis to the range of 50-70 C reduces the task to a simple analysis of the influence of temperature on a concrete structure with constant parameters. The temperature field and stress analysis of heavy-concrete transfer-purge chamber of nuclear power plant should also take into account temperatures higher than 70 oC, as some mechanical and thermal parameters of the material change.
I propose to show the experimental results how these parameters change in the range of 70 - 100 oC.
2. Conclusion are general in nature and should be supplemented by the results of the above tests.
3. Why did the authors assume that the temperature rise time is 48 hours?
Author Response
Dear reviewer,
Thank you very much for your review on 5 Nov. 2022 with which you sent us the yours suggestions on our paper with the reference number materials-2007156. We also wish to take this opportunity to thank you for his constructive comments and valuable recommendations. We have carefully revised the manuscript according to your suggestion. The following reply hopes to answer your review,and please see the attachment.
If the reply cannot explain your suggestion, I hope to get your guidance again, thank you very much !
Best wishes!
Wang xiaohui

Reviewer 3 Report
This paper deals with the steady and transient-state temperature fields and stresses of the transfer purge chambers of the nuclear power plant structure, under normal operation and after 48 h of abnormal operation. Analyses were conducted by using the Abaqus finite-element software and the directly coupled method.
There is no visible scientific contribution in the work. Existing software was used to analyze the temperature field and the stress field due to the expected temperature load for a specific building without general conclusions and recommendations related to improving the structure's response in terms of stress reduction. Also the basic theory related to heat transfer (second chapter) is poorly written with a lot of mistakes. Material properties and material designations (Chapter 3.1) are not harmonized in the text and Table 1. The work is more professional than scientific and, in my opinion, does not correspond to the scope of this journal.
Author Response
Dear reviewer,
Thank you very much for your review on 8 Nov. 2022 with which you sent us the yours suggestions on our paper with the reference number materials-2007156. We also wish to take this opportunity to thank you for his constructive comments and valuable recommendations. As mentioned in your suggestion, there is no visible scientific contribution in the work. It is only based on software was used to analyze the temperature field and the stress field due to the expected temperature load for a specific building without general conclusions and recommendations related to improving the structure's response in terms of stress reduction. Thank you very much for your kind suggestion. Due to the rough writing and vague language expression of my previous article, I apologize for the inconvenience caused to your review, and we have made the changes based on your suggestion,Please see the attachment.
With your suggestion, we found that we still have a lot of follow-up work to improve. I hope that after completing this part of the work, I will seriously consider your suggestion ,and start to conduct in-depth research on this research to make the data more detailed,and the scope of application is wider. The research results provide a reference for other similar structures. If the above reply cannot explain your advice, I hope to get your guidance again, thank you very much !
Best wishes!
Wang xiaohui

Round 2
Reviewer 2 Report
Paper has been supplemented to a satisfactory extent.
The work is understandable. The research is done in a relatively narrow scope, but it is done correctly.
I propose to publish the work in the current version.
Reviewer 3 Report
The authors did not successfully respond to comments and suggestions. The basic theory related to heat transfer and heat transfer modes is still wrong. It needs to be repaired. Also in chapter 3.1, the material properties and material types mentioned in the text and the table are still inconsistent.
Author Response
Dear reviewer,
Thank you very much for your review on 20 Dec. 2022 with which you sent us the yours suggestions on our paper with the reference number materials-2007156.
Since some writing problems with the first draft caused a lot of trouble for your review, We apologize again.
1. In the second part of the article--Basic theory of heat transfer, I revised the basic theory related to heat transfer, correcting wrong formula information. This part of the basic theory of heat may not be fully understood by me. If there is something wrong with this revision, I kindly ask the reviewers to help me revise it so that I can learn this aspect.
2. I've already changed Material properties and material designations (Chapter 3.1) ,and made them consistent in text and Table 1 .
I would like to take this opportunity to thank you for your second guidance and valuable advice. With your second guidance, I carefully read this book《Heat transfer》 and studied the knowledge of thermodynamics seriously. And I have carefully revised the manuscript according to your suggestion. Please guide again, if there is something wrong, I implore you to help me modify it, because the basic theory in this area may be difficult for me, I have not understood it. I request you can make correct revisions for me. And that I can learn this knowledge well. I hope to get your help and guidance again, thank you very much !
Thank you and best regards.
Yours sincerely,
Xiaohui Wang
Corresponding author:
Name: xiaohui Wang
E-mail: [email protected]

Round 3
Reviewer 3 Report
Sentence in line 107 and 108 is wrong. It should be:
In thermodynamics, there are three modes of heat transfer: conduction, convection and radiation.